# Clustering of Dietary Patterns Associated with Health-Related Quality of Life in Spanish Children and Adolescents

**DOI:** 10.3390/nu16142308

**Published:** 2024-07-18

**Authors:** José Francisco López-Gil, Mayra Fernanda Martínez-López

**Affiliations:** 1One Health Research Group, Universidad de Las Américas, Quito 170124, Ecuador; josefranciscolopezgil@gmail.com; 2Department of Communication and Education, Universidad Loyola Andalucía, 41704 Seville, Spain; 3Cancer Research Group, Faculty of Medicine, Universidad de Las Américas, Quito 170124, Ecuador

**Keywords:** eating healthy, food patterns, lifestyle, well-being, mental health, youth

## Abstract

Purpose: The aim of the current study was to examine the association between dietary patterns and health-related quality of life (HRQoL) among Spanish children and adolescents. Methods: A modified version of the parental version of 10 items of the Screening for and Promotion of Health-Related Quality of Life in Children and Adolescents—a European Public Health perspective (KIDSCREEN-10) was used to assess children’s HRQoL in three areas: subjective physical, mental, and social status. To evaluate dietary habits, a food frequency questionnaire was employed. To identify different feeding patterns in the sample of children and adolescents examined, cluster analyses were carried out. In addition, a generalized linear model with a Gaussian distribution was applied to test the associations between the determined clusters and HRQoL. Results: The lowest HRQoL was identified in participants located in the unhealthiest cluster (Cluster 1) (mean [M] = 85.2; 95% confidence interval [CI] 83.7 to 86.7). In comparison with the unhealthiest cluster (Cluster 1), a greater estimated marginal mean of HRQoL was identified for participants in the moderately healthy cluster (Cluster 1) (*p* = 0.020) and in the healthiest cluster (Cluster 2) (*p* = 0.044). Conclusions: Based on our findings, dietary habits based on the low consumption of bread, cereals, and dairy products (mainly), together with low intake of fruits and vegetables, are related to lower HRQoL in children and adolescents. These results underscore the importance of promoting balanced and nutrient-rich diets among young populations. Public health initiatives should focus on educating parents, caregivers, and children about the benefits of a varied diet that includes adequate portions of fruits, vegetables, whole grains, and dairy products.

## 1. Introduction

Health-related quality of life (HRQoL) is a complex concept that comprises an individual’s physical, emotional, and social well-being and functioning [1,2]. It is widely recognized as an important public health outcome [3] and is considered more than just an indicator of physical and mental health in young people [4]. Guaranteeing the well-being of young people is essential for healthy development, positive behaviors, and future success [5]. Moreover, higher HRQoL has been linked to greater life satisfaction [6], while lower HRQoL has been associated with higher odds of suicide-related behaviors in youth [7]. Therefore, identifying modifiable factors that can influence HRQoL in adolescents is critical for the development of interventions that can prevent or treat unseen health needs, which can have a significant impact on an individual’s overall well-being [6,7].

Regarding dietary patterns, the World Health Organization emphasizes that a healthy diet protects against malnutrition in all its forms and against noncommunicable diseases such as diabetes, cancer, stroke, and heart disease [8,9]. There is a global need to improve dietary habits, as a deficient diet is a greater risk factor for death than traditionally recognized factors such as smoking [10]. Supporting this perspective, providing evidence-based assistance for healthier dietary patterns and lifestyles can play a crucial role in public health [11]. Promoting nutritional education and awareness can empower individuals to make informed food choices, thereby enhancing overall well-being [12]. Additionally, implementing policies that support access to affordable, nutritious food options is essential in fostering healthier communities globally [13].

Previous studies have highlighted the link between healthy dietary patterns and HRQoL. For example, a systematic review by Vajdi and Farhangi [14] revealed that the Mediterranean dietary pattern and other “healthy” dietary patterns were related to higher HRQoL dimension scores in both mental and physical summaries, whereas “unhealthy” and “Western” dietary patterns were linked to lower HRQoL scores. Additionally, Victoria-Montesinos et al. [15] noted that adherence to the Mediterranean diet, particularly daily fruit consumption, was associated with higher HRQoL among Brazilian and Spanish preschoolers, children, and adolescents during the coronavirus disease 2019 (COVID-19) lockdown. Similarly, Shariati-Bafghi et al. [16] reported that a Mediterranean-style dietary pattern was linked to better HRQoL among healthy Iranian adolescents. Supporting this, Jiménez-López et al. [17] reported that higher adherence to the Mediterranean diet was related to higher HRQoL, independent of sociodemographic, physical, and lifestyle factors. Furthermore, through a cluster analysis, these same authors reported that a group with a greater proportion of participants skipping breakfast had significantly lower HRQoL scores.

Therefore, the aim of the current study was to examine the association between dietary patterns and HRQoL among Spanish children and adolescents. Evaluating HRQoL in young people is crucial, as those with low HRQoL during childhood are less likely to maintain a healthy path into adulthood [18]. Hence, identifying dietary patterns related to increased HRQoL in children and adolescents could offer guidance for the development of nutritional intervention programs, particularly at early ages.

## 2. Materials and Methods

### 2.1. Population and Study Design

This nationwide cross-sectional study utilized data from the 2017 Spanish National Health Survey, overseen by the Ministry of Health, Consumer Affairs, and Social Welfare, in collaboration with the National Statistics Institute [19,20]. The study employed a three-stage sampling methodology: the first stage involved selecting census sections, the second stage involved selecting households, and the third stage involved selecting individuals. From each household, one person aged 15 or older was chosen to complete the Adult Questionnaire, and, if the household included children aged 0–14 years, one child was randomly chosen to complete the Minor Questionnaire. Participants were informed of the survey process through a letter from the Ministry, which outlined the survey’s objectives, the voluntary and anonymous nature of participation, and the visit from an authorized and qualified interviewer.

This study specifically analyzed data from the Minor Questionnaire, targeting children aged 0–14 years. The initial sample comprised 6101 participants (100.0%). However, since the HRQoL assessment was only applicable to participants aged 8 years or older, 2875 participants (47.1%) younger than 8 years old were excluded. Additionally, 317 participants (5.2%) were removed because of incomplete dietary pattern data, and 201 participants (3.3%) were excluded due to missing information on various covariates (e.g., socioeconomic status, screen time, body mass index). Consequently, the final sample consisted of 2708 Spanish children and adolescents aged 8–14 years (44.3%). The complete process of sample selection is shown in Figure 1.

The data for this study were provided by the Ministry of Health, Consumer Affairs, and Social Welfare and were publicly available on the official Spanish Government website [19]. Ethical approval from a committee was not required according to Spanish law, as the data used were secondary and anonymized.

### 2.2. Procedures

#### 2.2.1. Health-Related Quality of Life

The Spanish National Health Survey made some changes to the parental version of 10 items of the Screening for and Promotion of Health-Related Quality of Life in Children and Adolescents – a European Public Health perspective (KIDSCREEN-10) to adapt it for the Eurobarometer study [19]. Specifically, question 7 was removed. The KIDSCREEN-10 assesses children’s HRQoL in three areas: subjective mental, physical, and social status. The questionnaire asked about situations experienced by the child or adolescent over the past week, including feeling good and in good shape, having a lot of energy, feeling sad or lonely, having enough time for themselves, being able to perform their desired activities in their free time, having fun with friends, doing well in school, and being able to pay attention. Responses were provided on a scale of “nothing”, “a little”, “moderately”, “very much”, and “a lot”. The modified KIDSCREEN-10 scores were converted into a scale of 0–100, with higher scores indicating better HRQoL. The primary and modified 9-item versions of the instrument have good reliability in European children, with Cronbach’s α values equal to or higher than 0.75 [21,22]. The Spanish version of the KIDSCREEN-10 has also been shown to be reliable and valid, with a Cronbach’s α of over 0.70 [23].

#### 2.2.2. Dietary Patterns

To evaluate dietary habits, a food frequency questionnaire was employed [19,20]. Parents or guardians were asked about the frequency with which their children consumed each food item. These food categories included “fruits”; “meat”; “eggs”; “fish”; “pasta, rice, potatoes”; “bread, cereals”; “vegetables”; “legumes”; “processed meat”; “dairy products”; “cookies, pastries, sweets, jams”; “sugar-sweetened beverages”; “fast food”; and “snacks”. Respondents were given six options to choose from: “never”, “less than once a week”, “once or twice weekly”, “three times weekly”, “four to six times weekly”, and “once or more daily”. For the purpose of converting the data into continuous variables, these categories were assigned scores as follows: “never” (zero points), “fewer than once a week” (one point), “once or twice weekly” (two points), “three times weekly” (three points), “four to six times weekly” (four points), and “once or more daily” (five points).

#### 2.2.3. Covariates

Parents or guardians supplied information about their child’s age, sex, and immigrant status (native-born or foreign-born). The socioeconomic status of the child was determined based on the occupation of the primary adult in the household. The physical activity levels of the child were assessed using a brief questionnaire adapted from the International Physical Activity Questionnaire [24], which focused on a single question regarding the child’s physical activity during leisure time. The response options were “no exercise” (sedentary activities such as reading, watching TV, going to the cinema, etc.); “occasional physical activity or sport”; “physical activity several times a month”; and “sports or physical training several times a week” [24]. The amount of recreational screen time was reported separately for weekdays and weekends with the following question: “How much time does your child typically spend on a weekday in front of a screen, including computers, tablets, TVs, videos, video games, or cell phones?” The response options were “no time or almost no time”, “less than one hour”, and “one hour or more”. The sleep duration of the child was assessed with the following question: “Approximately how many hours does your child usually sleep daily? (Including nap times)”.

### 2.3. Statistical Analysis

The data are provided in two forms: absolute and relative numbers (%) for categorical variables and means (M) and standard deviations (SDs) for continuous variables. Cluster analysis was conducted to determine different dietary patterns. Standardized scores (z-scores) of the consumption of each food group were computed and used as cluster inputs to ensure consistency. No sign inversion was performed according to the a priori healthy or unhealthy properties of the foods. Prior research in Spanish samples has identified three clusters a priori [17,25,26,27]. However, we confirmed this number of clusters using two methods. Firstly, a hierarchical cluster analysis with Ward’s method, according to squared Euclidean distances [28], was conducted to determine the appropriate number of clusters. Hierarchical cluster analysis using Ward’s method merges clusters to minimize the total within-cluster variance, measured by squared Euclidean distances. This process forms a dendrogram to visualize the hierarchical clustering structure. Secondly, k-means cluster analysis provided the final cluster solution and the number of clusters identified in the first step, resulting in the establishment of three clusters: Cluster 1 (the unhealthiest), Cluster 2 (moderately healthy), and Cluster 3 (the healthiest). The k-means clustering algorithm partitions data into “k” distinct clusters by minimizing the variance within each cluster. It iteratively assigns data points to the nearest centroid and updates the centroids until convergence, ensuring efficient and scalable pattern identification in large datasets. A detailed description of these clusters can be found in Appendix A. Additionally, a generalized linear model with a Gaussian distribution was used to test the associations between the determined clusters and HRQoL, while controlling for variables such as age, sex, immigrant status, socioeconomic status, physical activity, recreational screen time, and sleep duration. The statistical analyses were conducted using the R statistical software (version 4.4.0) developed by the R Core Team in Vienna, Austria and RStudio (2024.04.1 + 748) from Posit in Boston, MA, USA. A *p* value less than 0.050 was considered to indicate statistical significance. A *p*-value below 0.05 was established for statistical significance.

## 3. Results

Table 1 shows the characteristics of the study participants. Overall, the mean HRQoL was 87.3 points (SD = 11.1). The greatest proportion of participants with a low socioeconomic status was found in Cluster 3 (46.7%). The highest proportions of participants with the highest physical activity level and the longest overall screen time duration were found in Cluster 2 (moderately healthy) (sports or physical training several times a week: 31.4%; overall screen time: M = 141.1 min; SD = 78.4). The lowest sleep duration was identified in Cluster 1 (the unhealthiest group) (sleep duration: M = 8.9 h; SD = 1.0; body mass index: M = 19.7 kg/m^2^; SD = 9.4).

To aid interpretation, the z-scores of the different food groups included in the clustering process are displayed in Figure 2. In general, Cluster 1 (the unhealthiest) had the lowest z-scores for bread, cereals, and dairy. Cluster 3 (the healthiest) displayed the highest z-score for fruits and vegetables, and the lowest consumption of sugar-sweetened beverages, fast food, and snacks. Regarding Cluster 2 (moderately healthy), participants within this cluster reported the highest z-scores for meat, eggs, pasta, rice, potatoes, and processed meat.

Figure 3 illustrates the estimated marginal means of HRQoL according to membership in a particular cluster after adjusting for several covariates. The lowest HRQoL was identified in those participants in the unhealthiest cluster (Cluster 1) (M = 85.2; 95% CI 83.7 to 86.7). Compared to the unhealthiest cluster (Cluster 1), a greater estimated marginal mean of HRQoL was identified for participants in the moderately healthy cluster (Cluster 1) (*p* = 0.020) and in the healthiest cluster (Cluster 2) (*p* = 0.044).

## 4. Discussion

These findings revealed three different clusters based on the dietary patterns established, which showed different HRQoL values in a national sample of Spanish children and adolescents (aged 8–14 years). Overall, Cluster 3 (the healthiest) had the highest consumption of fruits, vegetables, pulses, and fish, and the lowest consumption of sugar-sweetened beverages, fast food, and snacks. This group exhibited greater HRQoL than did Cluster 1 (the unhealthiest). Furthermore, Cluster 2 (moderately healthy) showed the moderate consumption of fruits, vegetables, and fish but also included the higher consumption of pasta, rice, potatoes, and bread. Moreover, Cluster 2 also had the highest consumption of cookies, pastries, sweets, jams, sugar-sweetened beverages, fast food, and snacks. However, it also had higher quality of life than Cluster 1 (the unhealthiest). Lastly, Cluster 1 (the unhealthiest) reported the low consumption of bread, cereals, and dairy (among others), which resulted in the lowest HRQoL. Although the specific mechanisms by which certain dietary patterns could lead to a higher HRQoL have not been fully elucidated, there are potential reasons for these results.

One possible explanation for these findings could be related to the lower consumption of bread, cereals, and dairy products in Cluster 1. In this context, a previous study by Sari et al. [29] among female adolescents from Indonesia demonstrated a significant positive association between the intake of certain nutrients (carbohydrates, calcium) and adolescents’ HRQoL. In support of this idea, another study by Davison et al. [30] revealed that the frequent consumption of bread and dairy foods was associated with greater well-being among male adolescents in the United Kingdom. Regarding dairy products, the scientific literature has highlighted the potential positive influence of vitamin D on children’s mental health [31], which could impact HRQoL [32]. For carbohydrates, a previous study by Fabios et al. [33] revealed that a high carbohydrate quality index was related to a lower risk of micronutrient inadequacy in Spanish children.

Another possible explanation for these results could be the lower consumption of fruits and vegetables observed in Cluster 1 (the unhealthiest). Visser et al. [34] conducted a study among Brazilian children and adolescents and showed that those who consumed fewer than three servings of fruits and vegetables daily had lower mental well-being and HRQoL. Additionally, Davison et al. [30] reported that female adolescents in the United Kingdom had greater mental well-being and HRQoL. Visser et al. [34] also reported that Brazilian children and adolescents who consumed fruits and vegetables on less than five days a week had lower mental well-being and HRQoL. A longitudinal study by Juton et al. [35] reported that the baseline consumption of fruits and vegetables (among others) was associated with higher HRQoL at follow-up. The high consumption of fruits and vegetables can provide adequate intake of carbohydrates, vitamins, and antioxidants [36], which have been linked to neurotransmitter concentrations and synthesis and may play a role in mitochondrial energy processes [37]. Thus, it is not surprising that it has been suggested that increased dietary diversity and increased fruit and vegetable consumption could serve as an approach to increase HRQoL among youths [38].

Furthermore, the consumption of ultra-processed foods (i.e., fast food, sweets, or snacks) may also be related to our findings. Cluster 3 (the healthiest) exhibited the lowest consumption of sugar-sweetened beverages, fast food, and snacks and greater HRQoL than did Cluster 1 (the unhealthiest), which might partially explain this observation. This outcome aligns with the existing scientific literature [34,35,39,40]. For example, a longitudinal study by Juton et al. [35] revealed that the consumption of fast food, baked goods, pastries, and sweets (among other items) was inversely associated with HRQoL. Similarly, Qin et al. [39] reported that Chinese children and adolescents with greater consumption of sugar-sweetened beverages or fast food experienced significantly lower HRQoL. Additionally, a study by da Costa et al. [40] revealed that a higher processed food score was associated with lower HRQoL. In Brazil, another study revealed that children and adolescents who consumed fast food twice a week or more had lower HRQoL than did those who did not [34].

On the other hand, our results showed that Cluster 2 (moderately healthy) consumed more cookies, pastries, sweets, jams, sugar-sweetened beverages, fast food, and snacks than Cluster 1 (the unhealthiest) and had higher HRQoL, which contradicts what was expected. However, Cluster 2 also consumed more healthy foods, such as pulses, meat, eggs, pasta, rice, and potatoes. Some studies have suggested that there may be a compensatory effect where consuming a diet rich in healthy foods, such as fruits, vegetables, and whole grains, can mitigate the negative effects of unhealthy foods, such as those high in saturated fats, sugars, and sodium [41]. Thus, the difference in HRQoL between Cluster 2 and Cluster 1 may be related to the overall quality of their diets, which seems to be a critical factor in determining HRQoL [14]. A systematic review and meta-analysis by Wu et al. [42] and a systematic review by Romero-Robles et al. [43] both revealed positive associations between healthy diets and HRQoL in multiple domains of HRQoL, including physical, school, and emotional functioning, and psychosocial quality of life among children and adolescents. The Mediterranean diet, which is characterized by the use of olive oil as the primary dietary fat and the abundant consumption of seasonal fruits, vegetables, legumes, whole grains, and nuts, with low consumption of red and processed meats, ultra-processed food, sweets, confectionery, and pastries [44], has also been linked to higher HRQoL in children and adolescents [43].

Some limitations and strengths of this study should be acknowledged. First, due to the cross-sectional design, it was not possible to determine the direction of the associations observed, as well as causal relationships. In this sense, it has been suggested that this relationship could also occur in the opposite direction (i.e., participants with lower HRQoL might have lower adherence to the Mediterranean diet and lower consumption of fruits and vegetables [45]). Therefore, future longitudinal studies are needed to clarify the direction of this association. Second, excess weight was established based on weight and height measurements declared by parents or guardians, which could have introduced measurement errors. Additionally, the use of self-report questionnaires presents the risk of social desirability and recall biases. Furthermore, parental feeding practices, which could influence dietary patterns, were not assessed in this study [46]. On a positive note, the study analyzed the associations in a representative and large sample of children and adolescents from the whole country, providing robust evidence with high external validity.

## 5. Conclusions

Based on our findings, dietary habits based on the low consumption of bread, cereals, and dairy products (mainly), together with the low intake of fruits and vegetables, are related to lower HRQoL in children and adolescents. These results underscore the importance of promoting balanced and nutrient-rich diets among young populations. Public health initiatives should focus on educating parents, caregivers, and children about the benefits of a varied diet that includes adequate portions of fruits, vegetables, whole grains, and dairy products. Such educational programs could be integrated into school curricula and community health services to maximize their reach and impact.

## Figures and Tables

**Figure 1 nutrients-16-02308-f001:**
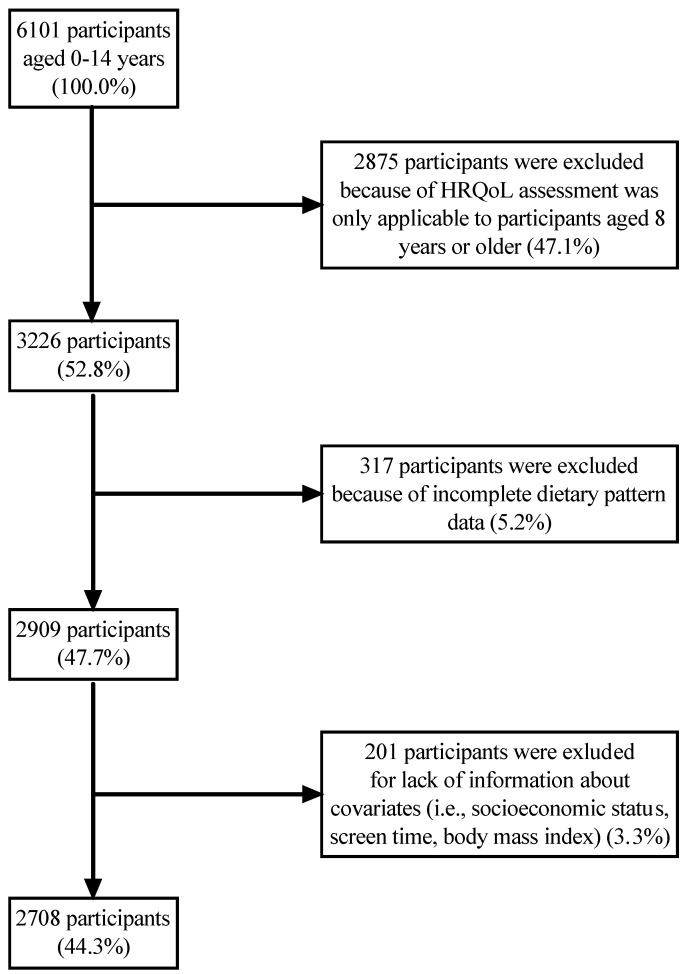
Flowchart depicting the selection of the study participants. HRQoL, health-related quality of life.

**Figure 2 nutrients-16-02308-f002:**
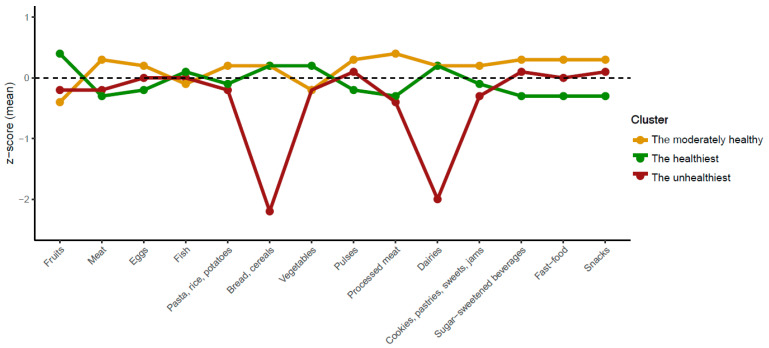
Standardized scores of different food consumption groups among Spanish children and adolescents with different established food consumption patterns. The colored lines indicate the z-score of each food group according to each established cluster.

**Figure 3 nutrients-16-02308-f003:**
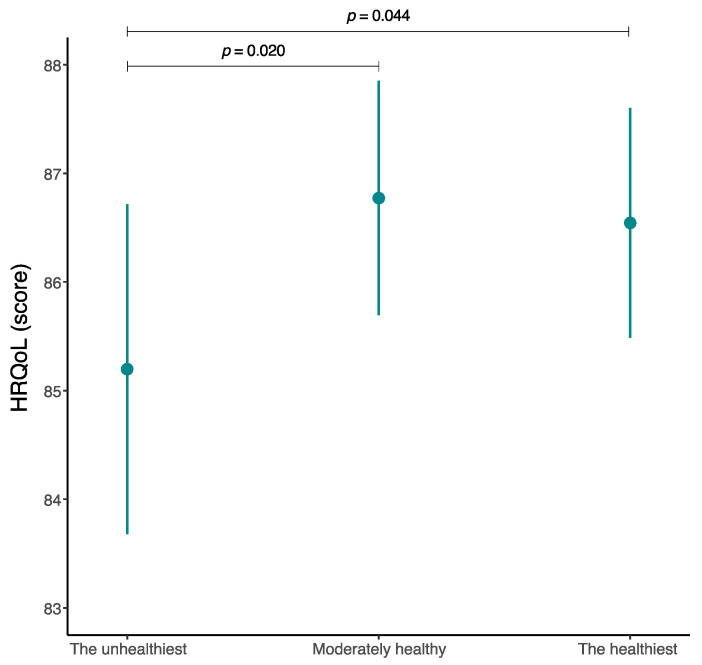
Estimated marginal means of health-related quality of life among Spanish children and adolescents according to the different established food consumption patterns. HRQoL, health-related quality of life.

**Table 1 nutrients-16-02308-t001:** Descriptive data of the study participants (overall and stratified by the clusters established) (*N* = 2708).

Variable		Cluster 1 (The Unhealthiest)	Cluster 2 (Moderately Healthy)	Cluster 3 (The Healthiest)	Total
Age	Mean (SD)	11.1 (2.0)	11.3 (2.0)	11.1 (2.0)	11.2 (2.0)
Sex	Boys (%)	88 (36.8)	619 (53.2)	636 (48.7)	1343 (49.6)
	Girls (%)	151 (63.2)	544 (46.8)	670 (51.3)	1365 (50.4)
SES status	Low SES (%)	103 (43.1)	574 (49.4)	566 (43.3)	1243 (45.9)
	Medium SES (%)	79 (33.1)	377 (32.4)	445 (34.1)	901 (33.3)
	High SES (%)	57 (23.8)	212 (18.2)	295 (22.6)	564 (20.8)
Immigrant status	Native-born (%)	231 (96.7)	1101 (94.7)	1252 (95.9)	2584 (95.4)
	Foreign-born (%)	8 (3.3)	62 (5.3)	54 (4.1)	124 (4.6)
Body mass index (kg/m^2^)	Mean (SD)	19.7 (4.5)	19.4 (3.6)	19.2 (3.6)	19.3 (3.7)
Physical activity (status)	No exercise (%)	37 (15.5)	157 (13.5)	172 (13.2)	366 (13.5)
	Occasional physical activity or sport (%)	46 (19.2)	231 (19.9)	252 (19.3)	529 (19.5)
	Physical activity several times a month (%)	75 (31.4)	349 (30.0)	412 (31.5)	836 (30.9)
	Sports or physical training several times a week (%)	81 (33.9)	426 (36.6)	470 (36.0)	977 (36.1)
Overall screen time (minutes)	Mean (SD)	130.2 (81.5)	141.1 (78.4)	117.6 (67.0)	128.8 (74.2)
Overall sleep duration (hours)	Mean (SD)	8.9 (1.0)	9.0 (1.0)	9.1 (1.0)	9.0 (1.0)
HRQoL (score)	Mean (SD)	85.7 (11.3)	87.5 (10.5)	87.4 (11.5)	87.3 (11.1)

HRQoL, health-related quality of life; SD, standard deviation; SES, socioeconomic status.

## Data Availability

The data used in this study are available upon request from the corresponding authors. However, given that the participants are minors, privacy and confidentiality must be respected.

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
