# Peer review of "Clustering of Dietary Patterns Associated with Health-Related Quality of Life in Spanish Children and Adolescents"

_nutrients, 2024, doi:10.3390/nu16142308_

Round 1
Reviewer 1 Report
Comments and Suggestions for Authors
Dr. López-Gil & Dr. Martínez-López report clustering them into three types of dietary habits using data from the 2017 Spanish National Health Survey youths and evaluated their relationship between their type and health status (HRQoL). The study shows that diet during growth influences future growth and health, and that nutritional education is necessary. This paper need some modifications.
1. Please add the legends of Figures 1 and 2, as they are missing. Also, please correct the labels on the axes in the figure as they are small and difficult to understand.
ï¼’. Please add a diagram of the selected sample data as a figure (P2, line 77~88).
3.The K-means clustering method is important and is explained in detail in METHODS. And also, Also, please illustrate the results in Figure.
Comments on the Quality of English LanguageThe level of English is not considered to be a problem.
Author Response
REVIEWER 1
Dr. López-Gil & Dr. Martínez-López report clustering them into three types of dietary habits using data from the 2017 Spanish National Health Survey youths and evaluated their relationship between their type and health status (HRQoL). The study shows that diet during growth influences future growth and health, and that nutritional education is necessary. This paper need some modifications.
Thank you so much for your time and feedback.
- Please add the legends of Figures 1 and 2, as they are missing. Also, please correct the labels on the axes in the figure as they are small and difficult to understand.
Done. Thank you.
ï¼’. Please add a diagram of the selected sample data as a figure (P2, line 77~88).
Done. Thank you.
3.The K-means clustering method is important and is explained in detail in METHODS. And also, Also, please illustrate the results in Figure.
Thank for your comment. Further information has been added about the cluster analyses procedures applied.

Reviewer 2 Report
Comments and Suggestions for Authors
This paper obtain a very important issue, but it contains some errors.
In the chapter “Dietary patterns” (2.2.2.0) it is said “…….For the purpose of converting the data into a continuous variable, these categories were assigned scores as follows: “never” (zero points), “fewer than once a week” (one point), “once or twice weekly” (2 points), “three times weekly” (three points), “four to six times weekly” (four points), and “once or more daily” (five points)”.
The same score was probably incorrectly assigned for the frequency of consumption for products considered both healthy and unhealthy. For this reason, for products such as “fruits”; “meat”; “eggs”; “fish”; “pasta, rice, potatoes”; “bread, cereals”; “vegetables”; “legumes”; “dairy products” a higher frequency of consumption should correspond to a higher number of points, while in the case of products such as “cookies, pastries, sweets, jams”; “sugar-sweetened beverages”; “fast food”; and “snacks” a higher frequency of consumption should correspond to a smaller number of points. In addition the maximum possible number of points and the values ​​obtained for the three selected feeding patterns were not provided. In this situation, it would be inappropriate to discuss the correlation between the Healthy Eating Index (score) and the HRQoL (score) index. Probably for this reason mentioned, the authors write that (line 243-245) “On the other hand, our results showed that Cluster 2 (moderately healthy) consumed 243 more cookies, pastries, sweets, jams, sugar-sweetened beverages, fast food, and snacks than Cluster 1 (the unhealthiest) and had a higher HRQoL, which contradicts what was expected”. Therefore, this work requires appropriate analysis and correction.
For this reason, other indicators, i.e. HRQoL and physical activity level, should also be checked for correct interpretation.
Author Response
This paper obtain a very important issue, but it contains some errors.
Thank you for your time and feedback.
In the chapter “Dietary patterns” (2.2.2.0) it is said “…….For the purpose of converting the data into a continuous variable, these categories were assigned scores as follows: “never” (zero points), “fewer than once a week” (one point), “once or twice weekly” (2 points), “three times weekly” (three points), “four to six times weekly” (four points), and “once or more daily” (five points)”.
The same score was probably incorrectly assigned for the frequency of consumption for products considered both healthy and unhealthy. For this reason, for products such as “fruits”; “meat”; “eggs”; “fish”; “pasta, rice, potatoes”; “bread, cereals”; “vegetables”; “legumes”; “dairy products” a higher frequency of consumption should correspond to a higher number of points, while in the case of products such as “cookies, pastries, sweets, jams”; “sugar-sweetened beverages”; “fast food”; and “snacks” a higher frequency of consumption should correspond to a smaller number of points. In addition the maximum possible number of points and the values ​​obtained for the three selected feeding patterns were not provided. In this situation, it would be inappropriate to discuss the correlation between the Healthy Eating Index (score) and the HRQoL (score) index. Probably for this reason mentioned, the authors write that (line 243-245) “On the other hand, our results showed that Cluster 2 (moderately healthy) consumed 243 more cookies, pastries, sweets, jams, sugar-sweetened beverages, fast food, and snacks than Cluster 1 (the unhealthiest) and had a higher HRQoL, which contradicts what was expected”. Therefore, this work requires appropriate analysis and correction.
For this reason, other indicators, i.e. HRQoL and physical activity level, should also be checked for correct interpretation.
Thank you for your comment. We understand the reviewer's point and this alternative interpretation. The reviewer suggests scoring the food patterns based on the "a priori" healthiness or unhealthiness of the food groups (i.e., from a qualitative perspective). In our case, the scoring is based on the quantity of food consumed (from a quantitative perspective). A higher score in "vegetables" means a higher intake of them, and a higher score in "sweets" means a higher intake of them. Generally, we consider it more appropriate to interpret food groups quantitatively (for analyses) rather than qualitatively. Some foods (for example, eggs) can be healthy when consumed in moderation and unhealthy when consumed in excess. Therefore, we believe that our quantitative analysis is appropriate and might be easier for readers to understand.

Round 2
Reviewer 2 Report
Comments and Suggestions for Authors
I am very sorry, but I cannot agree with the explanations of the autors.
In the original Healthy Eating Index (HEI) the points obtained are summed, and in the modified Spanish Healthy Eating Index (S-HEI) probably also?
For example,
“vegetables” eaten ”once or twice weekly” (2 points) '' and "fast food" eaten “once or more daily” (5 points). The sum is 7 points
And now it's the other way around
“vegetables” eaten “once or more daily” (5 points) ' and "fast food" eaten once or twice weekly” (2 points). The sum is 7 points too
Is it the same ?
Additionally, I have no answer to the following statement from the first review
“In addition the maximum possible number of points and the values ​​obtained for the three selected feeding patterns were not provided”.
In “Discussion” chapter it is said
“Cluster 3 (the healthiest), characterized by the highest consumption of fruits, vegetables, pulses, and fish, exhibited a greater HRQoL than did Cluster 1 (the unhealthiest). Furthermore, Cluster 2 (moderately healthy) showed moderate consumption of fruits, vegetables, and fish but also included higher consumption of pasta, rice, potatoes, and bread. Last, Cluster 1 (the unhealthiest) reported a
characterized low consumption of bread, cereals, and dairies (among others), which resulted in the lowest HRQoL”.
So why was the frequency of consumption of unhealthy products (“cookies, pastries, sweets, jams”; “sugar-sweetened beverages”; “fast food”; and “snacks”) assessed and not taken into account?
Author Response
Please, see the attachment.

Round 3
Reviewer 2 Report
Comments and Suggestions for Authors
Necessary explanations and additions have been made to the text